# Development and pilot test of a smartphone app for midwifery care in Tanzania: A comparative cross-sectional study

Yoko Shimpuku[1]*, Beatrice Mwilike[2], Dorkasi Mwakawanga[2], Keiko Ito[3], Naoki Hirose[1], Kazumi Kubota[4]

**1** Graduate School of Biomedical and Health Sciences, Hiroshima University, Hiroshima, Japan, **2** School of Nursing, Muhimbili University of Health and Allied Sciences, Dar es Salaam, Tanzania, **3** Kyoto University Hospital, Kyoto, Japan, **4** Department of Healthcare Information Management, The University of Tokyo Hospital, Tokyo, Japan

* yokoshim@hiroshima-u.ac.jp

## Abstract

To address Tanzania's high maternal mortality ratio, it is crucial to increase women's access to healthcare. To improve access, the quality of antenatal care needs to be improved. Therefore, we conducted a pilot study of a smartphone app for midwives and examined its potential effects on the learning outcomes of midwives and birth preparedness of pregnant women in Tanzania. This mixed-methods, pilot study provided an educational app for midwives in the intervention group, obtained data about the continuous use of the app, measured midwives' learning outcomes, directed focus group discussions on the usability of the app, and conducted surveys among pregnant women about birth preparedness in the intervention and control groups to evaluate if midwives provided proper information to them. The control group received regular antenatal care and answered the same survey. Participants were 23 midwives who participated in the testing and provided learning outcome data. Twenty-one participated in focus group discussions. Results showed that 87.5% of midwives continued to study with the app two months post-intervention. A mini-quiz conducted after using the app showed a significant increase in mean scores (6.9 and 8.4 points, respectively) and a non-significant increase on the questionnaire on women-centered care (98.6 and 102.2 points, respectively). In the focus group discussions, all midwives expressed satisfaction with the app for several reasons, including comprehensive content, feelings of confidence, and reciprocal communication. There were 207 pregnant women included in the analysis. The intervention group had significantly higher knowledge scores and home-based value scores than did controls. The total scores and other subscales did not show statistical significance for group differences. The results indicate the potential impact of the midwifery education app when it is implemented on a larger scale, especially considering that the results show a potential effect on midwives' learning outcomes.

**Data Availability Statement:** There are ethical restrictions against releasing our data transcripts publicly, because they contain sensitive information, and our study participants are from a

vulnerable population. Through the process of obtaining research permission by the research ethics committee, restrictions of data access were imposed. If researchers obtain additional permission, we will make the fully de-identified data available upon request to Hiroshima University via email (kasumi-kenkyu@office.hiroshima-u.ac.jp) or phone ((81) 82-551 257-5555).

**Funding:** The authors report the following sources of funding: Japan Society for the Promotion of Science awarded a grant (17K17486/20K10935), Kyoto University to YS, and Japan Agency for Medical Research and Development grant (22hk0302012h0201) awarded to YS. The funders had no role in study design, data collection and analysis, decision to publish, or preparation of the manuscript.

**Competing interests:** The authors have declared that no competing interests exist.

## Introduction

In low- and middle-income countries (LMICs), maternal and neonatal mortality are serious issues and are included in the United Nations Sustainable Development Goals [1]. Women need to have access to healthcare and preparation that begins when they become pregnant. However, pregnant women in sub-Saharan Africa continue to have low access to healthcare. In Tanzania, 51% of pregnant women have four or more antenatal care (ANC) visits [2], which is far below the recent World Health Organization (WHO) recommendation of at least eight visits during pregnancy [3].

Research suggests that there is a need for improving the quality of ANC in Tanzania [4, 5]. A cross-sectional descriptive study performed in 11 health facilities in Kigoma district stated that basic blood and urine tests, such as hemoglobin and urine albumin, were insufficient and substandard and that 20% of severe maternal morbidities were attributed to substandard ANC [4]. When investigating perceptions of ANC among clients and providers, researchers found that both clients and providers perceived that the experience of care was important to clients and providers according to the availability of physical and human resources and the content of the care delivered [5]. A study about provider adherence to first-visit ANC standards showed that after adjusting variables based on the level of health facilities, better adherence to first-visit ANC standards was significantly associated with the following variables: having female providers at dispensaries, the performance of quality assurance at a health center, the availability of routine tests and basic medicine at a hospital; the availability of medicine and receiving refresher training at public facilities, and receiving external funds from the government at a private facility [6]. To mitigate the time constraint that midwives experience when providing ANC and refresh their knowledge of the care they should provide, Oka et al. [7] showed that a job aid could help providers follow guidelines to provide information to pregnant women.

Similarly, Shimpuku et al. [8, 9] developed an antenatal education program using both WHO guidelines and their own research to locally contextualize the guidelines used to teach danger signs and birth preparation among pregnant women and their families in rural Tanzania. The program was originally developed as a picture drama (a paper-based aid comprising pictures and descriptive text) and used for health education in communities. The study found that the health worker educational aids had a potential effect on increasing birth preparedness and reducing maternal complications, such as bleeding, seizures, cesarean sections, and neonatal complications.

While conducting previous studies, we found extensive smartphone use in Tanzania, especially among the young population. Within the research team, which included researchers in Japan and Tanzania, communication became quick because of increase in smartphone use. In research, for example, the Basic Emergency Care course application was developed and applied in Tanzania. The researchers reported limitations in Internet access; however, they found potential utility [10]. Hence, the research team considered that the use of an educational app is possible, especially among professional groups with economic stability. Therefore, to expand the reach of the program to a larger population, this study developed a smartphone app with the goal of providing easy access to healthcare professionals anywhere in Tanzania.

There is a series of mHealth (mobile health using cell phones) studies on the prevention of mother-to-child transmission of HIV in Tanzania [11–13]. The researchers found some positive effects: an increase in the numbers of women testing positive for HIV compared with the pre-intervention period, good feasibility based on tablet-based surveillance [11], health worker capacity-building and patient reminders merged into a single robust and responsive system [12], and significant improvement from the pre-test for the total survey and questions concerning system attitudes [13]. According to a randomized control study of mHealth in Kenya

[14], three methods of breastfeeding support were compared: 1) one-way short message service (SMS), 2) two-way SMS with a nurse, and 3) control. They found that both one-way and two-way SMS were effective in improving exclusive breastfeeding practices, and that two-way SMS had an added benefit over one-way. Two-way SMS was used to support exclusive breastfeeding when challenges occurred and to discuss ways to overcome cultural pressures that can influence sustained breastfeeding through its promotion of two-way connections and exchanges. However, there are also challenges in mHealth interventions. For example, a systematic study on mHealth interventions in LMICs reported technical problems including network coverage, Internet access, electricity access, and maintenance of mobile phones [15]. A separate systematic review on mHealth interventions in ANC programs in sub-Saharan Africa mentioned the absence of supporting data for a scale up of such interventions, such as costing and cost-effectiveness data [16], while another systematic review for clinical decision support systems showed healthcare providers' concerns about increased workload and altered workflow hinders sustainability [17].

To pre-empt some of the known challenges with mHealth, the app developed in this study used Dar es Salaam as the target site, where the network availability is highest in the country. The team included the app developers, who helped when the participants needed technical support. As this study was at a foundational stage, there was additional workload for the healthcare providers; hence, they were compensated for their additional work of inputting data into the app. The research team further decided to embed reciprocal communication using a social media function.

Considering the current literature highlighting the possible value of educational programs and smartphone apps in ANC, this pilot study tested the app, especially on continuous use among midwives and its potential effects on 1) the learning outcomes of midwives and 2) birth preparedness of pregnant women in Tanzania.

## Materials and methods

### Design

This mixed-methods, pilot study evaluates a smartphone app including midwives (part 1) and pregnant women (part 2) in urban Tanzania (Fig 1). The purpose of the app was to provide updated information on WHO guidelines and practical suggestions for midwives to use for health education at ANC visits. In part 1, a pilot study was tested among midwives in the intervention group using: a) data about continuous use of the app, b) understanding of the contents and women-centered care, and c) focus group discussions (FGDs) about the usability of smartphones. There was no intervention for midwives in the control group. In part 2, surveys were conducted among pregnant women about birth preparedness in two groups (the intervention and control groups).

### Setting

The study was conducted in two health facilities in Dar es Salaam, which is the city with the largest population in Tanzania (population approximately 4.3 million) [18]. In urban areas, 63.8% of women had more than four ANC visits, and only 45% of women in rural areas had four or more visits. Women who received ANC from skilled providers in Dar es Salaam had the highest percentage: 35% for receiving ANC from doctors (including assistant medical officers, clinical officers, and assistant clinical officers) and 60.3% from nurses, midwives, and assistant nurses [2]. The two institutions were randomly assigned to either the intervention group or the control group. The study took place between October 2019 and March 2021.

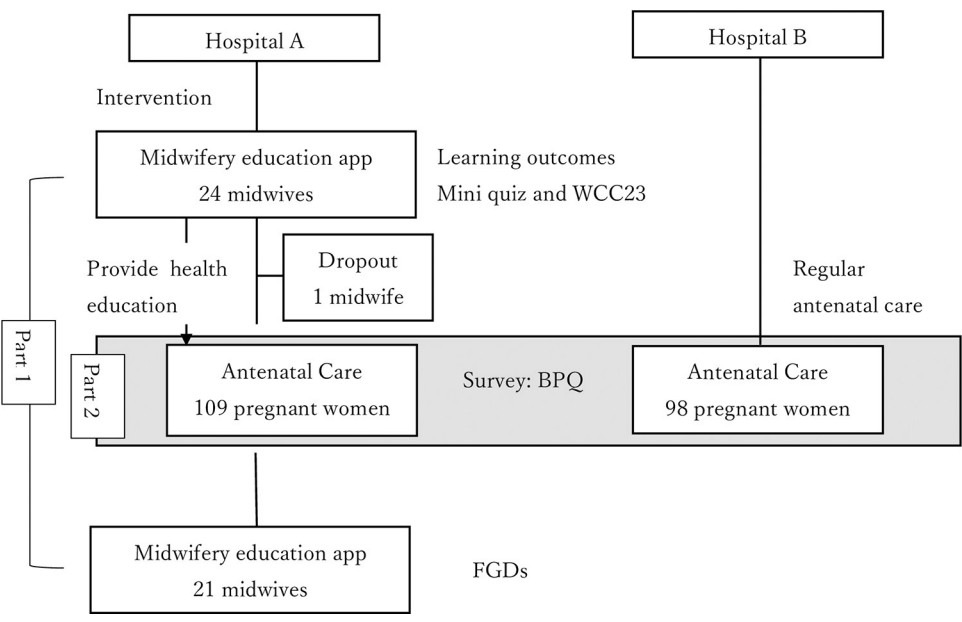

**Fig 1. Recruitment and data collection process.**

## Participants

The study included both midwives and pregnant women. The inclusion criteria for midwives were those 1) working in the antenatal ward of the intervention site, and 2) having a smartphone. The inclusion criteria for pregnant women were 1) being in the second trimester or later, 2) coming to the health facility for antenatal checkups, 3) being 16 years or older, and 4) being able to read Kiswahili. Midwives were recruited only in the intervention group because the control group received regular ANC. Pregnant women were recruited for both the intervention and control groups.

To compare the two groups of pregnant women, the sample size of this study was calculated based on the basic formula with two groups, a two-sided alternative and normal distribution with the same variances. The sample size was calculated as 64 for each group to detect a difference (10 points) between groups at a 5% level of significance with 80% power. Concerning 40% of the missing data, the minimum number of participants needed in each group was 90.

## Intervention

We followed the WHO guidelines for reporting mHealth [19] and summarized the related parts in Table 1. The infrastructure for smartphones in Tanzania has been increasing. Global Digital reported that there were 44.13 million mobile phone connections in Tanzania in January 2020, which was equivalent to 75% of the total population [20].

We selected an online education platform called Goocus (Castalia Co., Ltd.) as it allows educational content to be applied to multiple functions, such as video, pictures, narrations, and quizzes. Goocus can be accessed via a smartphone app or PC, and we used an app for the intervention. We converted the picture materials from a previous study [9, 11] into a video with narrations in Kiswahili, as most local women understand that language better than English. Along with the WHO recommendations on antenatal and intrapartum care for positive pregnancy and childbirth experiences [3, 21], we created contents with locally adapted illustrations that explained why preventive behaviors or early treatment were important and

**Table 1. Notes based on the mHealth evidence reporting and assessment guideline by WHO.**

| Criteria | Item no | Notes from the study | Page no where item is reported |
|---|---|---|---|
| Infrastructure (population level) | 1 | Global digital reported that there were 44.13 million mobile phone connections in Tanzania in January 2020, which was equivalent to 75% of the total population. | 7 (can be replaced with actual page number on publication) |
| Technologyplatform | 2 | We selected an online education platform called Goocus (Castalia, co. ltd.) as it allows educational contents to be applied to multiple functions, such as video, pictures, narrations, quizzes, etc. | 7 |
| Interoperability/Health information systems (HIS) context | 3 | The system is not integrated into existing health information system as the app provides an educational content, rather than collecting health information. | 8 |
| Intervention delivery | 4 | The Goocus can be accessed in the form of a smartphone app or PC, and we used an app for the intervention. | 7–8 |
| Intervention content | 5 | Along with the WHO recommendations on antenatal and intrapartum care for positive pregnancy and childbirth experiences, we created contents with localized illustrations that explained why the preventive behaviors or early treatment were important, and how midwives could reflect them in their daily care. | 8 |
| Usability/content testing | 6 | This study is formative research to test the feasibility. | Overall study |
| User feedback | 7 | User feedback was summarized in Table 2 based on the qualitative analysis. | Table 2 |
| Access of individual participants | 8 | Barriers and facilitators of the adoption of the intervention among study participants were also described in Table 2 and discussion of the study. | Table 2 Discussion |
| Cost assessment | 9 | They were also provided the reimbursement (10,000 Tanzania Shillings) for the data they need to use the application. | 9 |
| Adoption inputs/programme entry | 10 | As a one-day training, midwives in the intervention site were instructed to download the application and how to use it by second and third authors. | 8–9 |
| Limitations for delivery at scale | 11 | Limitation was also described in Table 2 and discussion of the study. | Table 2 Discussion |
| Contextual adaptability | 12 | We use English and Kiswahili as the main languages of the app, but it can be translated into any other languages. | 8 |
| Replicability | 13 | The methods section explains the details of intervention so that they study could be replicated. | Methods |
| Data security | 14 | The app was secured by the passcord so that only the developers and researchers had access to the data. | 10 |
| Compliance with national guidelines or regulatory statutes | 15 | As the study used the WHO guidelines of antenatal and intrapartum care, it is aligned with national/regulatory guidelines of Tanzania. | 8 |
| Fidelity of the intervention | 16 | This study is a feasibility study to show the fidelity. | Overall study |

how midwives could demonstrate them in their ANC. A practical "ANC by week guide" was created so that midwives could find and check the number of gestational weeks of pregnant women during antenatal checkups. We also created a mini-quiz based on the content for the midwives to check their understanding of the educational content and determine what they would need to review. In addition, each item has a space for comments and "likes" to allow midwives to communicate with each other and with the developers. The system is not integrated into existing health information systems, as the app provides educational content rather than collecting health information. The app included both English and Kiswahili texts. As the study used the WHO guidelines for antenatal and intrapartum care, it is aligned with the national and regulatory guidelines of Tanzania.

**Ethical approval and consent to participate.** The study was conducted based on the principles of ethics, such as harmlessness, voluntarily, anonymity, and protection of privacy and personal information. These principles were explained to participants during recruitment. The research team explained the purpose, methods, and ethical considerations and asked each participant (both midwives and pregnant women) if they agreed to participate in the study. Only those who agreed to participate and provided written consent were included in the study. Ethical clearance and permissions were obtained from: 1) the Kyoto University Graduate School

and Faculty of Medicine, Ethics Committee (C1446), 2) the National Institute for Medical Research, Tanzania (NIMR/HQ/R.8/Vol.IX/1604), and 3) the Tanzania Commission for Science and Technology (2013–273-NA-2013-101).

## Data collection process

**Site assignment.** The data collection process is illustrated in Fig 1. Two health facilities were selected, as they were similar in size, patient number, and accessibility. They were randomly assigned to either an intervention or a control group. The second author allocated the numbers to the institutions, and the first author conducted allocation using a computer random number generator without knowing which institution was allocated to which number. Hence, allocation was conducted with concealment for the researchers.

**Training and use of the app among midwives.** As a one-day training, midwives in the intervention group were instructed to download the app and were shown how to use it by the second and third authors. They were also provided with reimbursement (10 000 Tanzania shillings; approx. 5 USD) to account for the mobile data they would use with the app as Wi-Fi is not widely available in Tanzania. Further, needing to pay for the mobile data might have hindered their use of the app. The researchers monitored their use of the app on Goocus and evaluated their usage two months after the intervention. When a midwife made comments on the app, the first author acted as a facilitator, responding to the comments, and asking probing questions of those who used it. Midwives in the control group did not receive the app and provided regular ANC for pregnant women.

**Focus group discussions among midwives in the intervention group.** All 23 midwives who used the app were approached again, and 21 participated in the FGDs, which were conducted twice with 10 or 11 members each. The second and third authors acted as facilitators of the discussion, recorded the discussion with permission, and checked transcription and translation from English to Kiswahili. Each session lasted approximately one hour.

**Exit survey among pregnant women.** One month after the intervention was administered, 212 pregnant women (110 in the intervention group and 102 in the control group) at the two health facilities were approached at the antenatal clinic at both facilities and explained the purpose and contents of the study. Only those who agreed to participate completed the questionnaire, using a tablet with an electronic form.

## Measurements

**Part 1: For midwives.** *App usage data*. As the app can record usage data, such as frequency of use, type of content accessed, and length of time on app, midwives' usage of the app was monitored and evaluated two months after they started using the app. The app was secured by a password so that only the developers and researchers had access to the data.

*Mini-quiz*. The quiz was developed based on the app content and was completed by the midwives. The quiz was composed of 10 multiple-choice questions from which one of four items could be chosen. Each correct answer was counted as 1 point, giving a range of 0 to 10 points.

*Women-centered Care Questionnaire*. The midwives completed the Women-Centered Care Questionnaire (WCC23E), which is a 23-item self-administered questionnaire that was developed based on the 50-item WCC-preg [22]. The original questionnaire aimed to measure women's perceptions of the care that they received during pregnancy. In the study by Horiuchi et al. [23], the questionnaire was used to measure the understanding of this concept among midwives. There are four factors associated with WCC: 1) being supported in making decisions, 2) having effective interaction, 3) being respected, and 4) trusting the caregiver.

The questionnaire items asked the healthcare providers if they agreed or disagreed with each item using a five-point Likert-type scale: 1) *strongly disagree*, 2) *somewhat disagree*, 3) *neither disagree nor agree*, 4) *somewhat agree*, and 5) *strongly agree*. Possible scores range from 23 to 115 points. The higher the score, the more healthcare providers felt that women-centered care was appropriately provided. The Cronbach's alpha coefficient for all 23 items was 0.835, indicating acceptable internal consistency.

*Focus group discussions*. FGDs were conducted to assess the midwives' perception of the app's usability based on a semi-structured interview guide. The two FGDs were conducted in Kiswahili with a Tanzanian midwifery researcher (third author) and recorded with permission and were then transcribed and translated into English. As all of the available participants in part 1 were approached and the discussion was conducted until new ideas stopped emerging, we concluded that data saturation was achieved. The English transcription was coded, and the content analysis was conducted by the first author.

**Part 2: For pregnant women.** *Birth Preparedness Questionnaire*. Pregnant women completed the Birth Preparedness Questionnaire (BPQ). The BPQ is a 34-item self-administered questionnaire consisting of a knowledge test and a birth preparedness assessment [9]. The 10-item knowledge test asked about safe pregnancy and danger signs derived from the Integrated Management of Pregnancy and Childbirth [24]. The responses were *yes* or *no*, and the score was calculated as 1 point per correct answer, giving a maximum score of 10. Another 24-item birth prepared assessment test is composed of items related to psychological values and beliefs. The items were rated using a three-point Likert scale indicating (1) *disagree*, (2) *neither disagree nor agree*, or (3) *agree*, with a maximum score of 72. Factors included home-based values (seven items), birth preparedness (five items), family support (four items), avoidance of medical intervention (two items), provision of money and food (two items), preference for skilled birth attendants (SBA) (two items), and pregnant women's workload (two items). The higher the score, the more the person understood the WHO recommendations in terms of birth preparedness and birth attended by skilled personnel. The questionnaire was administered in Kiswahili because it is familiar to most Tanzanians. The research assistants interviewed the participants and collected the data electronically using tablets. Cronbach's alphas of all factors were reported as home-based value (0.846), birth preparedness (0.691), family support (0.646), avoidance of medical intervention (0.548), preparation of money and food (0.615), preference of SBA (0.472), and pregnant women's workload (0.337).

## Analysis

**Statistical methods.** To compare the score of the mini-quiz and WCC23E in midwives before and after intervention, we conducted a Wilcoxon rank sum test.

To compare differences in background data between the two groups, an independent t-test was used for numerical data (i.e., age and parity) and a Chi-square test was used for categorical data. For cesarean section and experience of baby loss, we only included multipara women because primipara women were out of the scope of these questions. To compare mean differences of the impact on midwives, such as the mini-quiz and WCC23E, a Wilcoxon signed rank test was used as the sample size was small.

To compare the score of each item from the 24-item BPQ (total score without knowledge scores, knowledge scores, home-based values, birth preparedness, family support, avoidance of medical intervention, provision of money and food, preference for SBA, and pregnant women's workload) between intervention and control groups, unadjusted and adjusted analysis of covariance (ANCOVA) were conducted. In the adjusted ANCOVA, we adjusted for parity

because it is reasonable to consider that parity would affect birth preparedness and the intervention group had significantly higher parity.

The multiple imputation method was applied to multiple regression analyses to complement the missing values. The 10 imputed datasets were applied with multivariate imputations by chained equations. The estimates from the 10 imputed datasets were combined with Rubin's rules for combining multiply imputed data [25]. All tests were two-tailed, and the threshold of significance was $P < .05$. All statistical analyses were two-tailed and were performed using R Version 3.0.1 and Oracle® R Enterprise, Version 1.4.1 (Oracle, Redwood Shores, CA, USA).

**Qualitative analysis.**  The translated transcription was reviewed by the first author, who conducted content analysis, retrieved codes from the texts, and merged codes into subcategories and categories. Garrard's Matrix Method [26] was used to overview and compare the texts. The contents were more abstract as they were merged into the upper level. The third author, who moderated the FGD, cross-checked and validated the analyzed result.

## Results

### Background information

**Part 1: Midwives.**  Within the intervention group, 24 midwives met the inclusion criteria and downloaded the app, and 23 provided background information. The average age was 42.96 years (range: 27–59). The educational background was secondary school (14, 61%), university (8, 35%), and primary school (1, 4%). For nursing/midwifery qualification, 10 participants had a certification (43%), 10 had a diploma (43%), and 3 had a university degree (13%). The average employment time was 14.74 years (range: 5–33). All participants regularly worked in antenatal clinics.

**Pilot test of the app.**  Among the 23 midwives, 87.5% continued to study with the app two months after the intervention. A total of 62.5% of participants completed the study module. The usage patterns are shown in Fig 2. Each line shows the patterns of how fast the participants proceeded with the modules, demonstrating that there were midwives who quickly completed them and those who gradually completed them. With regards to the mini-quiz, there was increase in the mean score before and after using the app (6.9 and 8.4 points, SD1.79 and 1.19, respectively; p = .018; Fig 3). Regarding the WCC23E, there was no significant difference; however, the scores increased, and the standard deviation decreased (98.6 and 102.2 points, SD 12.05 and 5,53, respectively; p = .39; Fig 4).

**Focus group discussions.**  Table 2 illustrates the categories and subcategories that emerged from the FGDs. The midwives described the effects of the app: it "provides useful knowledge for practice" and "increases confidence and ability." The knowledge midwives mentioned included access to healthcare, preparation for birth, danger signs, services at antenatal visits by weeks, care during childbirth, nutrition during pregnancy, safety issues during pregnancy, provision of information to families, and information related to COVID-19. One midwife said,

> "This app has helped me to know the mother's preparation before she gives birth: she needs to prepare her family and leave people to look after her family. And it has also helped me to know those danger signs for a pregnant woman and what precautions to take, [and] when she sees them at home, she should rush to the hospital." (FGD 1, No. 9).

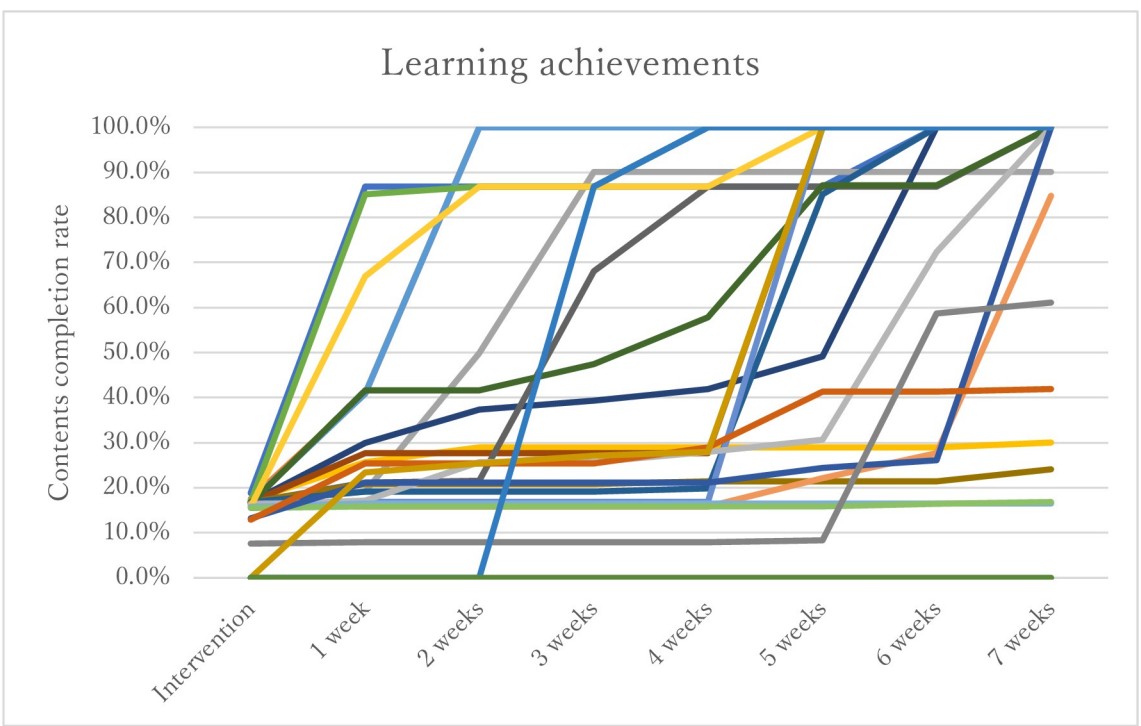

**Fig 2. Learning achievements of midwives.**

The app also increased confidence and ability to educate pregnant women through the use of app because the midwives felt that they had the correct knowledge at hand.

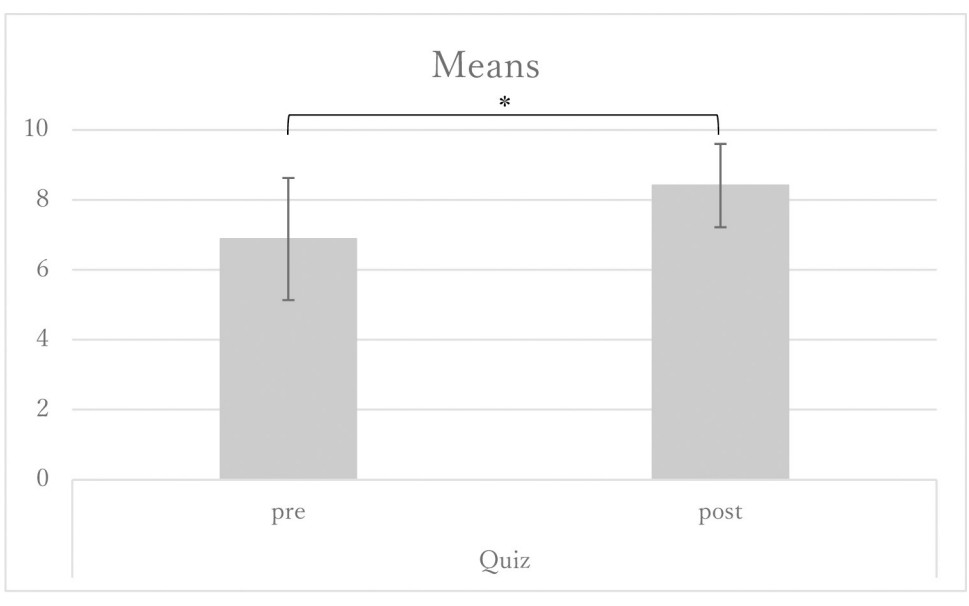

\* p < 0.05

**Fig 3. Means of the mini-quiz, pre- and post-intervention.**

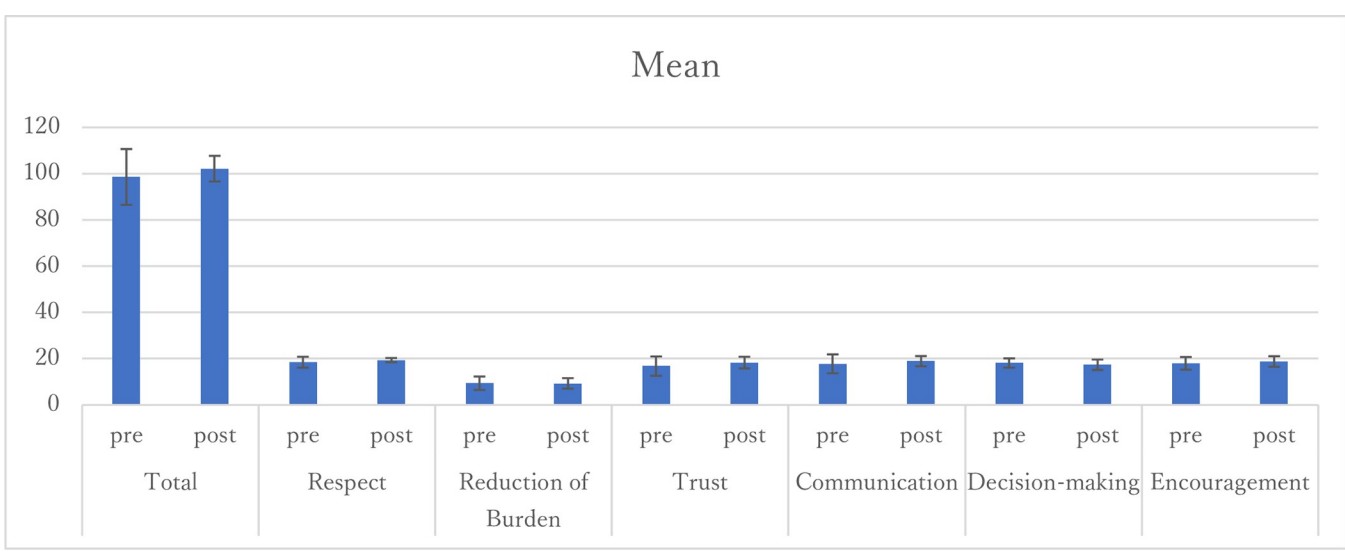

**Fig 4. Means of WCC23E, pre- and post-intervention.**

**Table 2. Categories and sub-categories emerged from the focus group discussion.**

| Categories | Sub-categories | Codes |
|---|---|---|
| Effects of the app | Provide useful knowledge for practice | It reminds me of the knowledge |
| | | New knowledge can be gained and updated |
| | Increases confidence and ability | It's fun because it gives me confidence |
| | | It increases my motivation to learn |
| | | It increases my skills |
| | | It tells me to follow the guidelines |
| Characteristics of the app | Easy to understand and use | The language used is easy to understand. |
| | | Easy to understand for mothers |
| | | It can be used anywhere |
| | Fun game elements | Others answered my questions |
| | | The quiz was like a game |
| | Usefulness of the contents | Comprehensive, from pregnancy to delivery |
| | | Safety is covered in many aspects |
| Further development of the app | More people to use it | I'd like you to send this to other midwives |
| | | I want to play the movie in the ward |
| | | Men should be educated, too |
| | More contents to be added | More information on childcare |
| | | Other infectious diseases, Ebola, Hepatitis. |
| | | About Vaccines |
| Improvements to the app | Improving environment | Put it in an electronic medical record |
| | | Need free internet access |
| | | Not enough time to use |
| | | Mothers forget what we taught |
| | Improving the app | I want to listen to the contents |
| | | I want more Swahili contents |
| | | I want to share the contents |
| | | Log-in problems |

"I found the app to be really good, and at first, it helped me at any time or anywhere I thought I could educate a pregnant woman; I educated her with confidence because I had material in my hand. So, I was always at peace even when I was in our village on holiday, I was able to open the app if I had pregnant women, and I continued providing education without any problems." (FGD 2, No. 5).

The characteristics of the app were evaluated as follows: "easy to understand and use" in terms of the language used and nutritional guidance tailored to the local diet; "fun" in terms of interactive and immediate communication both domestically and internationally and "game elements," such as quizzes. A midwife said,

". . .And when I faced a question, I am thankful that there were people who were very active, especially that white woman [the first author], she was active sometimes even at night. When you asked a question, she answered quickly. So, in short, I enjoyed that I had material in my hand all the time." (FGD 2, No. 5).

The app was also evaluated for its "usefulness of content," which is comprehensive from pregnancy to delivery and promotes safety for both medical professionals and pregnant women.

"I think they need to get it because our main goal is to help this mother get her baby safe and get her out safely. So, if they get it, it will help reduce maternal and child mortality." (FGD 2, No. 7).

In the development of the app, they recommended that "more people use it," including other midwives, pregnant women, and husbands and "more content be added," such as information on childcare, infectious diseases including Ebola and hepatitis, and vaccines. They also brought up "improving environment" which meant, for example, that since the smartphone app was a new method, using a smartphone in a clinical area could be misinterpreted.

"Our colleague says that you are with the client, and she sees that you are on the phone, and she feels as if you are ignoring her. [She would think] You are chatting [on the smartphone] (laughter)" (FGD 2, No. 10)

Other examples include overuse of bundles, as free Internet access is generally not available in the midwives' environment. It was also mentioned that the midwives had limited time to look at the app because of their daily work, and that pregnant women may forget what they have been told.

"Sometimes you do not get time to read because of the busyness of the work schedule that we have, we have many clients. Most of the time, I was reading when I am on the bus on my way home. You tried to read, but because of tiredness you find yourself asleep, maybe you targeted 10 questions, but every day you end up on the same question 6. So, the app itself, I can say it was not a challenge, but the challenges were on us due to a lot of work and a lot of customers." (FGD 1, No. 10).

To improve the app, they recommended that the reading materials be changed into voice recordings to that they could listen to the contents rather than reading them. They wanted more Kiswahili content, as they were familiar with the language. They wanted a share function,

so they could share content with other midwives. There were some problems in log-in because of forgotten passwords.

**Part 2: Pregnant women.**    A total of 207 pregnant women (109 in the intervention group and 98 in the control group) were included in the analysis. Demographic information is summarized in Table 3. There were significant differences in means of age and parity between the groups. The intervention group included older women with higher parities. There were significant differences in multipara women concerning cesarean section and experience of baby loss. There were no significant differences between the groups concerning marital status, education status, occupation, financial status, household assets, distance from health facility, number of ANC visits, first ANC visit, decision maker for birthplace, and preference of birth attendants.

**Birth Preparedness Questionnaire.**    Table 4 shows the comparison of BPQ scores between the intervention and control groups in unadjusted and adjusted ANCOVA. The intervention group had higher knowledge scores (p = .048) and home-based value scores (p = .033) as compared to control. The total scores and other subscales did not show statistical significance for group differences. The findings did not differ even after adjustment for parity and showed statistical significance in knowledge (p = .033) and home-based value score (p = .016).

## Discussion

### Pilot test of the app

With the app usage and FGDs, it is plausible to say that the piloting of the app was successful with free Internet access because of the high continuing rate, increase in mini-quiz scores, and positive comments from midwives. Based on the FGDs, all of the midwives expressed their satisfaction with the app, with important elements including comprehensive contents, feelings of confidence, and reciprocal communication of the app. Similarly, Birkmeyer et al. [27] conducted a literature review on determinants of mHealth success and identified user satisfaction as a key determinant of intention to continue use, and user satisfaction is created through personalization, interaction, mobile app design, and social networking. Another recent literature review on app intervention also revealed a promising effect on patients' physical and mental health [28]. Their feasibility and usability were also evaluated as effective; however, they found a similar limitation in the lack of Internet connection. It is necessary to install Wi-Fi in the institution when the app is officially introduced. One interesting aspect from the FGDs was that midwives stated that they used the app during their transportation to and from the workplace. Online training at the institution could resolve the issue of the Internet; however, as midwives usually live far from the city center, where the health institution is generally located, it is convenient for them to use the app during their travel. Hence, it would be important to discuss with the app developers if the app contents could be downloaded when they have Internet access and then used offline. Other obstacles such as the app conflicting with operating systems and battery failure owing to high data usage were not found, as our app was developed with an existing platform to avoid these issues.

In the review, one study that found that the app was not effective attributed this finding to the app's lack of interactive features (the intervention was podcasts). Although the app in this study was not created for healthcare providers rather than for patients or pregnant women, the results overlapped with the mHealth studies for patients. Regardless, it is important to consider how users interact with content and with other users.

It is necessary to arrange the environment so that midwives feel comfortable using the app, such as using a poster to inform patients that midwives may use their smartphone for their job. It is also key to prepare stable and free Internet access. In a scoping review of telehealth during COVID-19 [29], the authors noted that no study was published from Africa on this

**Table 3. Demographic characteristics of the pregnant women.**

| | Control | Inervention | P-value |
|---|---|---|---|
| n | 98 | 109 | |
| Age (mean (SD)) | 25.80 (4.45) | 28.94 (5.64) | <0.001 |
| Parity (mean (SD)) | 0.65 (0.96) | 1.21 (1.28) | 0.001 |
| Marital status (%) | | | 0.514 |
| Single | 24 (24.5) | 24 (22.0) | |
| Others | 73 (74.5) | 85 (78.0) | |
| NA | 1 (1.0) | 0 (0.0) | |
| Education status (%) | | | 0.568 |
| < College | 75 (76.5) | 85 (78.0) | |
| ≧ College | 22 (22.4) | 24 (22.0) | |
| NA | 1 (1.0) | 0 (0.0) | |
| Occupation (%) | | | 0.287 |
| House wife | 48 (49.0) | 42 (38.5) | |
| Others | 13 (13.3) | 20 (18.3) | |
| NA | 37 (37.8) | 47 (43.1) | |
| Financial status (%) | | | 0.069 |
| ≦ 5000TSH | 40 (40.8) | 28 (25.7) | |
| > 5000TSH | 50 (51.0) | 70 (64.2) | |
| NA | 8 (8.2) | 11 (10.1) | |
| Household asset (%) | | | 0.054 |
| Nothing | 27 (27.6) | 17 (15.6) | |
| ≥ one | 71 (72.4) | 92 (84.4) | |
| Caesarean section (%) | | | 0.001 |
| No | 27 (27.6) | 56 (51.4) | |
| Yes | 12 (12.2) | 13 (11.9) | |
| NA | 59 (60.2) | 40 (36.7) | |
| Experience of baby loss (%) | | | 0.003 |
| No | 32 (32.7) | 58 (53.2) | |
| Yes | 7 (7.1) | 11 (10.1) | |
| NA | 59 (60.2) | 40 (36.7) | |
| Distance (%) | | | 0.306 |
| ≦ 1 hour | 63 (64.3) | 69 (63.3) | |
| > 1 hour | 33 (33.7) | 40 (36.7) | |
| NA | 2 (2.0) | 0 (0.0) | |
| Number of antenatal care (%) | | | 0.675 |
| ≦ 3 | 22 (22.4) | 23 (21.1) | |
| ≧ 4 | 70 (71.4) | 82 (75.2) | |
| NA | 6 (6.1) | 4 (3.7) | |
| First antenatal care visit (%) | | | 0.345 |
| ≦ 3 months | 53 (54.1) | 48 (44.0) | |
| > 3 months | 44 (44.9) | 60 (55.0) | |
| NA | 1 (1.0) | 1 (0.9) | |
| Decision maker (%) | | | 0.508 |
| Only pregnant women | 73 (74.5) | 76 (69.7) | |
| Others | 20 (20.4) | 23 (21.1) | |
| NA | 5 (5.1) | 10 (9.2) | |
| Delivery attendants (%) | | | 0.553 |

(*Continued*)

**Table 3.** (Continued)

|  | Control | Inervention | P-value |
|---|---|---|---|
| n | **98** | **109** |  |
| Health care provider | 76 (77.6) | 87 (79.8) |  |
| No health care provider | 22 (22.4) | 21 (19.3) |  |
| NA | 0 (0.0) | 1 (0.9) |  |

topic. Other low- to upper-middle-income countries already have some telehealth-related infrastructure in place. The authors stated that it is important to establish the feasibility and utility of telemedicine in resource-limited settings so that the benefits are not missed. In Africa, it is more plausible to use a smartphone app than telehealth, which is why it was used as the platform in this study, and this study's findings provide necessary preparation for mHealth interventions in resource-limited settings.

## Learning outcomes of midwives

Although our findings were partially significant, we found a tendency of increasing knowledge and WCC23E scores among midwives and found the potential for an app approach in midwifery education. A review of digital, social, and mobile technologies (mobile phones, apps, tablets, Facebook, Twitter, and YouTube) in health professional education revealed that of the studies that reported evaluative outcomes (49.6%), most suggest that learners across all levels are typically satisfied with the use of the technologies for their learning [30]. Another scoping review of studies on the use of digital technology in undergraduate and/or postgraduate education in occupational therapy and/or physiotherapy reported that a wide variety of digital technologies were used, including quizzes, videos, social media, learning management systems, and content repositories [31]. The authors concluded that technology should not be used in isolation and must be aligned with the proposed learning outcomes and face-to-face contacts. The procedure and results of this study support this, as we used the app to improve the quality of care, and we provided in-person training. Starting the intervention with clear instructions for midwives was necessary, as studying with the app was new for them.

## Impact on pregnant women

In this study, we used the BPQ for outcome evaluation. However, the impact was limited; only knowledge and home-based values showed significant changes. One possible explanation is the mismatch between the intervention and outcome evaluation. In the future, another scale could be used to evaluate the impact of the app in another study. As we aim to improve the quality of care for women, we might want to use WCC23E for outcome evaluation in a future study. In addition, we provided a smartphone midwifery education and evaluated its impact on pregnant women. Building on the current research, another version of an app for pregnant women would be developed that allowed them to directly receive the impact of the intervention. As the midwives also stated in the FGDs that women forget the information they provided, it is better for women to have information on hand to review at any time.

To further discuss the results of the BPQ, it is interesting to find that only knowledge and a subscale of home-based value showed significant differences between the groups. Using this app, it is easier to increase knowledge on subjects such as danger signs and birth preparedness. Several studies have shown an increase in knowledge through mHealth interventions [32–34]. A literature review of mHealth interventions targeting pregnancy intakes in LMICs also showed the effect of mHealth intervention, which was in the form of a simple SMS or voice

**Table 4. The comparison of BPQ scores between the intervention and control groups in unadjusted and adjusted ANCOVA.**

| Group | Number | Total without k score | | K score total | | H score total | | B score total | | F score total | | A score total | | M score total | | S score total | | W score total | |
|---|---|---|---|---|---|---|---|---|---|---|---|---|---|---|---|---|---|---|---|
| | | Estimated mean | SE | Estimated mean | SE | Estimated mean | SE | Estimated mean | SE | Estimated mean | SE | Estimated mean | SE | Estimated mean | SE | Estimated mean | SE | Estimated mean | SE |
| Unadjusted | | | | | | | | | | | | | | | | | | | |
| Control | 98 | 57.5 | 3.8 | 9.4 | 0.94 | 19.6 | 2.43 | 12.3 | 1.22 | 10.6 | 1.59 | 4.02 | 0.54 | 5.7 | 0.68 | 4.1 | 0.50 | 5.2 | 1.08 |
| Intervention | 109 | 58.1 | 3.1 | 9.7 | 0.66 | 20.3 | 1.67 | 12.5 | 0.87 | 10.3 | 1.67 | 3.93 | 0.49 | 5.8 | 0.51 | 4.1 | 0.63 | 5.2 | 1.23 |
| P-value | | 0.2 | | < 0.05* | | 0.03* | | 0.11 | | 0.13 | | 0.19 | | 0.11 | | 0.41 | | 0.84 | |
| Adjusted | | | | | | | | | | | | | | | | | | | |
| Control | 98 | 57.5 | 3.8 | 9.4 | 0.94 | 19.6 | 2.43 | 12.3 | 1.22 | 10.6 | 1.59 | 4.02 | 0.54 | 5.7 | 0.68 | 4.1 | 0.50 | 5.2 | 1.08 |
| Intervention | 109 | 58.1 | 3.1 | 9.7 | 0.66 | 20.3 | 1.67 | 12.5 | 0.87 | 10.3 | 1.67 | 3.93 | 0.49 | 5.8 | 0.51 | 4.1 | 0.63 | 5.2 | 1.23 |
| P-value | | 0.15 | | 0.03* | | 0.02* | | 0.27 | | 0.36 | | 0.17 | | 0.12 | | 0.62 | | 0.99 | |

*: P-value < 0.05; SE: standard error

K score: Knowlede score; H score: Home-based value score; B score: Birth preparedness; F score: Family support score; A score: Avoidance of medical intervention score; M score: Provision of money and food score; S score: Preference of SBA score; W score: Pregnant women's workload score; Marital status: Others include married, divorced, and widowed; Occupation: Others include business, telor, teacher, specialist, farmer, assistant, media, student.

message, as a supplement to increase adherence [35]. Hence, when simple knowledge was being conveyed, the impact of mHealth interventions is easier to find.

The increase in the subscale of home-based value means that pregnant women in the intervention group had changed their minds and increased their preference for giving birth at health facilities. It could be that the improvement in the health education of midwives became an opportunity to build trust with pregnant women. Generally, in Africa, studies have reported negative results regarding the quality of healthcare and trust with healthcare providers. For example, in Ghana, several studies reported avoidance of SBA at health facilities owing to insensitivity or lack of knowledge about their culture [36–38]. In contrast, in rural Zambia, women preferred traditional birth attendants because they perceived them to be respectful, skilled, friendly, trustworthy, and available when they needed them [39]. These characteristics were found among midwives in developed countries. For example, in Australia [40], midwives create a trusting relationship with women and call it the "with woman" model. The authors stated that in this model the trusting relationship is the conduit for being "with woman" which influences midwifes and the profession of midwifery, as well as women and their families. Davis et al. [41] discussed how midwives are expected to look busy in hospital settings, compared with the environment of home or a birth center where midwives were able to engage in "being with" women. They showed that workplace culture impacted midwives' practices and recommended that all birthplace settings be conducive to midwifery practice.

Other subscales of the BPQ might have been difficult to shift during the intervention because they were more likely to be influenced by the conditions of the household, including monetary preparation for birth, providing nutritious food for pregnant women, and not working hard during pregnancy. Education with picture dramas in the previous study was more effective in increasing birth preparedness [9]; the result might have been owing to the involvement of family members in the intervention. It is again better to prepare an app for pregnant women so that they can share education with family members at home. A review study that created a theoretical model of mHealth also illustrated that if the intervention modality of mHealth provides positive psychological support or perceived satisfaction, women could improve health-seeking behaviors, whereas if it provides perceived information overload, it could decrease healthcare facility visits [42]. Therefore, the information that women receive should be simple and easy to share with families when they wish.

A limitation of this study was the small sample size and lack of baseline data, as it was a pilot study. In addition, the BPQ scale is limited as the reliability of some of its subscales are not high. Future studies should be conducted on a larger scale and follow pregnant women before and after intervention. Additionally, they should use other scales for outcome measurement. We could not use allocation concealment owing to the nature of the study design, which may cause some bias on the outcome. One possible influence of COVID-19 in this study was that the data collection was delayed in the control group owing to the pandemic; therefore, the pregnant women in the control group were affected by the situation with COVID-19. As it was unexpected for the pandemic to have occurred during the data collection period, its interruption was unavoidable; however, it may be beneficial to collect data from both the intervention and control groups concurrently so that the difference in time would not affect the data.

## Conclusion

This pilot study indicates the potential impact of the midwifery education app when it is implemented on a larger scale, especially considering that the results show a potential effect on midwives' learning outcomes. A future study with a larger sample should use women's perceptions of women-centered care as an outcome evaluation of the app intervention, and women should

also receive a simple app that reminds them of the contents of health education and shares that content with family members.

## Acknowledgments

We are grateful to the participants in Tanzania for giving up their time to answer the questionnaires. Our sincere gratitude goes to Dr. Sebalda Leshabari, who supported this study and arranged for younger researchers to contribute to the study. We thank Miss Minami Suzuki and CEO Satoshi Yamawaki from Castalia Co., Ltd. for development of the app. We also thank Editage for English language editing.

## Author Contributions

**Conceptualization:** Yoko Shimpuku, Beatrice Mwilike, Kazumi Kubota.

**Data curation:** Yoko Shimpuku, Beatrice Mwilike, Dorkasi Mwakawanga, Keiko Ito.

**Formal analysis:** Yoko Shimpuku, Naoki Hirose, Kazumi Kubota.

**Funding acquisition:** Yoko Shimpuku.

**Investigation:** Yoko Shimpuku, Beatrice Mwilike, Dorkasi Mwakawanga, Keiko Ito, Kazumi Kubota.

**Methodology:** Yoko Shimpuku, Naoki Hirose, Kazumi Kubota.

**Project administration:** Yoko Shimpuku, Beatrice Mwilike, Dorkasi Mwakawanga, Keiko Ito.

**Resources:** Yoko Shimpuku, Beatrice Mwilike, Keiko Ito.

**Software:** Naoki Hirose, Kazumi Kubota.

**Supervision:** Yoko Shimpuku.

**Validation:** Beatrice Mwilike, Dorkasi Mwakawanga, Naoki Hirose, Kazumi Kubota.

**Visualization:** Yoko Shimpuku, Naoki Hirose.

**Writing – original draft:** Yoko Shimpuku, Naoki Hirose.

**Writing – review & editing:** Yoko Shimpuku, Beatrice Mwilike, Dorkasi Mwakawanga, Keiko Ito, Naoki Hirose, Kazumi Kubota.

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
