## [Decision Letter · Decision Letter 0]

21 Sep 2022

PONE-D-22-03266Development and feasibility test of a smartphone app for midwifery care in Tanzania: A comparative cross-sectional studyPLOS ONE

Dear Dr. Shimpuku,

Thank you for submitting your manuscript to PLOS ONE. After careful consideration, we feel that it has merit but does not fully meet PLOS ONE’s publication criteria as it currently stands. Therefore, we invite you to submit a revised version of the manuscript that addresses the points raised during the review process.

I apologize for the delay in issuing a decision regarding this manuscript. It has been very difficult to secure reviewers. I have included my detailed review below to supplement the feedback provided by Reviewer #1.

We look forward to receiving your revised manuscript.

Kind regards,

Michelle L. Munro-Kramer, PhD, CNM, FNP-BC

Academic Editor

PLOS ONE

Journal Requirements:

3. Please include the following request in the decision letter, and ping me (lverduci@plos.org) with follow up. “Please include a complete copy of PLOS’ questionnaire on inclusivity in global research in your revised manuscript. Our policy for research in this area aims to improve transparency in the reporting of research performed outside of researchers’ own country or community. The policy applies to researchers who have travelled to a different country to conduct research, research with Indigenous populations or their lands, and research on cultural artefacts. The questionnaire can also be requested at the journal’s discretion for any other submissions, even if these conditions are not met.  Please find more information on the policy and a link to download a blank copy of the questionnaire here: https://journals.plos.org/plosone/s/best-practices-in-research-reporting. Please upload a completed version of your questionnaire as Supporting Information when you resubmit your manuscript.

Additional Editor Comments (if provided):

Thank you for submitting this manuscript on "Development and feasibility test of a smartphone app for midwifery care in Tanzania: A comparative cross-sectional study". While this is an important topic, there are a number of areas in this manuscript that need further explanation before it is suitable for publication. The biggest issue is that it is described as a feasibility test focused on feasibility and usability in the purpose; however, the methods have a heavy focus on proposed outcomes and then the discussion focuses on the feasibility and acceptability (page 25, line 413-415). There needs to be consistency among the purpose, methods, and discussion. Is this truly a pilot feasibility study focused on feasibility, acceptability, and usability with measurement of some proposed outcomes for future scale-up? I would recommend paying careful attention to Reviewer #1's comments as well as addressing the following:

1) INTRODUCTION: The section on mHealth needs to be expanded. Why is this a suitable approach for this problem? What are some of the limitations related to continuing education for healthcare providers in LMICs? What has the literature found in regard to limitations (e.g., failure to scale-up mHealth interventions) and how will this study address or build on these limitations? I would specifically look at the following references:

a) Amoakoh-Coleman M, Borgstein AB, Sondaal SF, Grobbee DE, Miltenburg AS, Verwijs M, Ansah EK, Browne JL, Klipstein-Grobusch K

Effectiveness of mHealth Interventions Targeting Health Care Workers to Improve Pregnancy Outcomes in Low- and Middle-Income Countries: A Systematic Review

J Med Internet Res 2016;18(8):e226.

b) Manyati, T. K., & Mutsau, M. (2021). A systematic review of the factors that hinder the scale up of mobile health technologies in antenatal care programmes in sub-Saharan Africa. African Journal of Science, Technology, Innovation and Development, 13(1), 125-131.

c) Adepoju, I. O. O., Albersen, B. J. A., De Brouwere, V., van Roosmalen, J., & Zweekhorst, M. (2017). mHealth for clinical decision-making in sub-Saharan Africa: a scoping review. JMIR mHealth and uHealth, 5(3), e7185.

2) METHODS: Please move the paragraph on ethical approval and consent to after the intervention section on page 10. The methods section has very little emphasis on capturing content related to feasibility and usability, but instead seems to capture proposed outcomes. Please explain this discrepancy if this is truly a pilot feasibility study. Additional questions that should be addressed:

a) Were patients responsible for completing the exit survey independently on the electronic form? What language was it in? What if they needed assistance? Why do you believe the reliability of some of the measures was so low?

b) Page 12, What type of app usage data (paradata) was gathered? Frequency of use, type of content accessed, length of time on app?

c) There are a few usability tools (e.g., The System Usability Scale by Brook, 1986) that expand beyond the paradata to capture other dimension of use. Why were these not used? It seems these types of items may have been included in the focus groups but there is little discussion of the topics/questions discussed in the focus groups. Please expand on this issue.

d) Page 16, line 154 should be "Chi-square"

e) Page 17, line 271-272, please cite the reference for Rubin's rules.

3) DISCUSSION: I would recommend caution in considering the app feasible based on proposed outcomes. As noted by Reviewer #1, if users cannot continue to use the app because of lack of wi-fi and data usage costs, then it may not be feasible. Instead it appears the results demonstrated that with free access the app is acceptable and usable.

Reviewers' comments:

Reviewer's Responses to Questions

**Comments to the Author**

1. Is the manuscript technically sound, and do the data support the conclusions?

Reviewer #1: Partly

2. Has the statistical analysis been performed appropriately and rigorously? 

Reviewer #1: Yes

3. Have the authors made all data underlying the findings in their manuscript fully available?

Reviewer #1: Yes

4. Is the manuscript presented in an intelligible fashion and written in standard English?

Reviewer #1: Yes

5. Review Comments to the Author

Reviewer #1: The manuscript covers a topic of high importance and the authors have done a very good job of considering relevant factors for intervention to improve access to Antenatal care for Tanzanian women. There are a number of limitations to the background and reporting of results that should be addressed prior to publication.

In the background section, the research presented that identifies specific factors that are associated with increased access to ANC is particularly helpful. What is missing then, is information that could elucidate the rationale behind choosing to move forward with an app to address some of these factors. Also missing in the background section are data that would support the choice of an app for the intervention delivery. Why an app rather than an educational program that could be accessed at work? If the authors are to support their claim that this program is feasible, they need to share data that will demonstrate widespread accessibility of apps, capacity to access and use them, and to do so at low cost. Also, why work with current midwives instead of trainees?

The methods are well described and results are clear and well presented.

A potentially major flaw is the consideration that this intervention is feasible. Although the data suggest a high degree of acceptability and use among midwives and mothers, this is in a context where the data fees to use the system in an environment where wifi is not prevalent. There is a substantive disconnect between making a system available free of charge for use and then considering whether and how you could encourage use when fees for data would be introduced. It remains unclear how or why people would use the app when fees for data are not covered.

The challenge here is that the content appears integral to the modality for delivery. The content for educational purposes is strong and the results suggest delivery of this content to midwives and pregnant women can be impactful. However, because content delivery is assumed to be feasible via an app a priori, the findings become highly problematic given the concerns about use when fees are in place as noted above. If the authors can separate out the findings to emphasize the value of delivering the content and then consider what modality--e.g., app versus online training program in the clinic--this would be an improvement. I am not suggesting re-doing the trial, but rather than the authors be much more circumspect about the choice of using an app to delivery this content. They should be more frank and forthcoming about the limitations of doing so and offer in the discussion section a critical consideration of other options for intervention delivery that might be more practical and feasible in this context.

6. PLOS authors have the option to publish the peer review history of their article (what does this mean?). If published, this will include your full peer review and any attached files.

Reviewer #1: **Yes: **Sheana Bull

---

## [Author Response · Author response to Decision Letter 0]

5 Nov 2022

Responses to the academic editor and reviewer(s)

We express our appreciation to the reviewers for their insightful comments, which helped us improve our study. We address your comments with point-by-point responses below, including quotations from the manuscript by page and line number. In the manuscript, we intended to denote changes through tracked changes, but an English editor delete authors’ changes. Hence, authors’ changes are highlighted in red.

Academic Editor: Major Comments

Comment: 1) INTRODUCTION: The section on mHealth needs to be expanded. Why is this a suitable approach for this problem? What are some of the limitations related to continuing education for healthcare providers in LMICs? What has the literature found in regard to limitations (e.g., failure to scale-up mHealth interventions) and how will this study address or build on these limitations? I would specifically look at the following references:

a) Amoakoh-Coleman M, Borgstein AB, Sondaal SF, Grobbee DE, Miltenburg AS, Verwijs M, Ansah EK, Browne JL, Klipstein-Grobusch K

Effectiveness of mHealth Interventions Targeting Health Care Workers to Improve Pregnancy Outcomes in Low- and Middle-Income Countries: A Systematic Review

J Med Internet Res 2016;18(8):e226.

b) Manyati, T. K., & Mutsau, M. (2021). A systematic review of the factors that hinder the scale up of mobile health technologies in antenatal care programmes in sub-Saharan Africa. African Journal of Science, Technology, Innovation and Development, 13(1), 125-131.

c) Adepoju, I. O. O., Albersen, B. J. A., De Brouwere, V., van Roosmalen, J., & Zweekhorst, M. (2017). mHealth for clinical decision-making in sub-Saharan Africa: a scoping review. JMIR mHealth and uHealth, 5(3), e7185.

Response: Thank you very much for your clarification and kind suggestions of literature to cite. We incorporated this feedback into the revised Introduction as follows:

“However, there are also challenges in mHealth interventions. For example, a systematic study on mHealth interventions in LMICs reported technical problems including network coverage, Internet access, electricity access, and maintenance of mobile phones [15]. A separate systematic review on mHealth interventions in ANC programs in sub-Saharan Africa mentioned the absence of supporting data for a scale up of such interventions, such as costing and cost-effectiveness data [16], while another systematic review for clinical decision support systems showed healthcare providers’ concerns about increased workload and altered workflow hinders sustainability [17].” (pp. 6–7, lines 131–142)

Comment: 2) METHODS: Please move the paragraph on ethical approval and consent to after the intervention section on page 10. The methods section has very little emphasis on capturing content related to feasibility and usability, but instead seems to capture proposed outcomes. Please explain this discrepancy if this is truly a pilot feasibility study.

Response: Thank you for highlighting this discrepancy and helping us improve the conveyance of our intended meaning. We moved this section, per your instructions. (pp. 11–12, lines 230–243)

We admit that we used the term “feasibility test” without clarifying our definition of the term. Upon referencing Arain et al.’s (2010)* study, we concluded that our study is a report of a pilot study, rather than a feasibility study. Hence, we revised our title as follows:

“Development and pilot test of a smartphone app for midwifery care in Tanzania: A comparative cross-sectional study.” (p. 1, lines 1–3)

We also revised relevant references from “feasibility” to “piloting” throughout the manuscript. We also revised the abstract accordingly.

“The results indicate the potential impact of the midwifery education app when it is implemented on a larger scale, especially considering that the results show a potential effect on the midwives’ learning outcomes.” (p. 3, lines 54–56)

*Arain M, Campbell MJ, Cooper CL, Lancaster GA. What is a pilot or feasibility study? A review of current practice and editorial policy. BMC Med Res Methodol. 2010;10: 67. doi: 10.1186/1471-2288-10-67.

Comment: Additional questions that should be addressed:

a) Were patients responsible for completing the exit survey independently on the electronic form? What language was it in? What if they needed assistance? Why do you believe the reliability of some of the measures was so low?

Response: We appreciate your inquiry. As the research assistants assisted women to answer questions, we revised the sentence as follows.

“The research assistants interviewed the participants and collected the data electronically using tablets.” (p. 17, lines 362–363)

Furthermore, because of the above assistance, we did not consider that these factors affected the reliability. The reliability of some of the measurements was low because the measurement, such as Avoidance of Medical Intervention and Preference of SBA, could be influenced by their experience of healthcare providers. Although there is limitation, we used the BPQ scale as this scale was developed for measuring birth preparedness in the context of Tanzania. Based on your comment, we added the following sentence in the limitation.

“In addition, the BPQ scale is limited as the reliability of some of its subscales are not high.” (p. 31, lines 669–670)

Comment: a) i) The language was written on the measurement section. 

Response: Thank you for noting our specification of the local language. We mentioned the language several times, Kiswahili, because it is the most widely used language by Tanzanian women in the context. Hence, we do not consider it to have affected reliability.

“We converted the picture materials from a previous study [9,11] into a video with narrations in Kiswahili, as most local women understand that language better than English.” (p. 10, lines 212–214)

Comment: b) Page 12, What type of app usage data (paradata) was gathered? Frequency of use, type of content accessed, length of time on app?

Response: Thank you for this insightful inquiry. The app can gather data on frequency of use, type of content accessed, and length of time on app. Hence, we revised as follows:

“As the app can record usage data, such as frequency of use, type of content accessed, and length of time on app, midwives’ usage of the app was monitored and evaluated two months after they started using the app.” (p. 14, lines 290–292)

Comment: c) There are a few usability tools (e.g., The System Usability Scale by Brook, 1986) that expand beyond the paradata to capture other dimension of use. Why were these not used? It seems these types of items may have been included in the focus groups but there is little discussion of the topics/questions discussed in the focus groups. Please expand on this issue.

Response: Thank you very much for your suggestion. As this was a pilot study, we wanted to test not only the app but also recruitment and measurement. I (the first author) am not an expert on the system and was not aware of the suggested tools. Now, we think that we could use the suggested tools in another study to deeply scrutinize the system itself. However, this is not the main purpose of the current study.

Comment: d) Page 16, line 154 should be "Chi-square"

Response: Thank you for the suggestion. We corrected this typo (p. 17, line 372).

Comment: e) Page 17, line 271-272, please cite the reference for Rubin's rules.

Response: Thank you for the suggestion. We included the citation accordingly (p. 18, line 393).

Rubin DB. Multiple Imputation for Nonresponse in Surveys. New York: Wiley; 1987. 258 p.

Comment: 3) DISCUSSION: I would recommend caution in considering the app feasible based on proposed outcomes. As noted by Reviewer #1, if users cannot continue to use the app because of lack of wi-fi and data usage costs, then it may not be feasible. Instead, it appears the results demonstrated that with free access the app is acceptable and usable.

Response: Thank you very much for your apt insight and suggestion. We agree that our descriptions did not indicate consideration of the limitations. We therefore revised the Discussion as follows:

“With the app usage and FGDs, it is plausible to say that the piloting of the app was successful with free Internet access because of the high continuing rate, increase in mini-quiz scores, and positive comments from midwives.” (p. 25, lines 541–543)

“It is necessary to install Wi-Fi in the institution when the app is officially introduced.” (p. 26, lines 561–562)

Reviewer #1’s comments: Major Comments

Comment:

a) The manuscript covers a topic of high importance and the authors have done a very good job of considering relevant factors for intervention to improve access to Antenatal care for Tanzanian women. There are a number of limitations to the background and reporting of results that should be addressed prior to publication.

In the background section, the research presented that identifies specific factors that are associated with increased access to ANC is particularly helpful. What is missing then, is information that could elucidate the rationale behind choosing to move forward with an app to address some of these factors. Also missing in the background section are data that would support the choice of an app for the intervention delivery. Why an app rather than an educational program that could be accessed at work? If the authors are to support their claim that this program is feasible, they need to share data that will demonstrate widespread accessibility of apps, capacity to access and use them, and to do so at low cost. Also, why work with current midwives instead of trainees?

Response: Thank you very much for the very constructive comment. We added the following explanation:

“While conducting previous studies, we found extensive smartphone use in Tanzania, especially among the young population. Within the research team, which included researchers in Japan and Tanzania, communication became quick because of increase in smartphone use. In research, for example, the Basic Emergency Care course application was developed and applied in Tanzania. The researchers reported limitations in Internet access; however, they found potential utility [10]. Hence, the research team considered that the use of an educational app is possible, especially among professional groups with economic stability” (p. 5, lines 104–111)

Comment: b) A potentially major flaw is the consideration that this intervention is feasible. Although the data suggest a high degree of acceptability and use among midwives and mothers, this is in a context where the data fees to use the system in an environment where wifi is not prevalent. There is a substantive disconnect between making a system available free of charge for use and then considering whether and how you could encourage use when fees for data would be introduced. It remains unclear how or why people would use the app when fees for data are not covered.

Response: Thank you very much for your valuable suggestion. We agree with your point and revised the Discussion as follows.

“With the app usage and FGDs, it is plausible to say that the piloting of the app was successful with free Internet access because of the high continuing rate, increase in mini-quiz scores, and positive comments from midwives.” (p. 25, lines 541–543)

“It is necessary to install Wi-Fi in the institution when the app is officially introduced.” (p. 26, lines 561–562)

Comment: c) The challenge here is that the content appears integral to the modality for delivery. The content for educational purposes is strong and the results suggest delivery of this content to midwives and pregnant women can be impactful. However, because content delivery is assumed to be feasible via an app a priori, the findings become highly problematic given the concerns about use when fees are in place as noted above. If the authors can separate out the findings to emphasize the value of delivering the content and then consider what modality--e.g., app versus online training program in the clinic--this would be an improvement. I am not suggesting re-doing the trial, but rather than the authors be much more circumspect about the choice of using an app to delivery this content. They should be more frank and forthcoming about the limitations of doing so and offer in the discussion section a critical consideration of other options for intervention delivery that might be more practical and feasible in this context.

Response: Thank you very much for your suggestions. We agree that we did not mention enough about the limitation regarding the fee of using the Internet. We did not mean that the result was applicable in real-world settings; rather, we meant that a larger study informed by our findings is possible and could be impactful. One point we would like to raise is that midwives used the app during their travels, as they usually live one hour or more away from the city center in Dar es Salaam because it is costly to live there. Hence, they want to go home early once they finish their work. Therefore, we did not choose online training at the institution as it would burden the midwives more. We had not mentioned these facts previously; thus, we added the following:

“It is necessary to install Wi-Fi in the institution when the app is officially introduced. One interesting aspect from the FGDs was that midwives stated that they used the app during their transportation to and from the workplace. Online training at the institution could resolve the issue of the Internet; however, as midwives usually live far from the city center, where the health institution is generally located, it is convenient for them to use the app during their travel. Hence, it would be important to discuss with the app developers if the app contents could be downloaded when they have the Internet access and then used offline.” (p. 26, lines 561–568)

We also revised our Conclusion based on your suggestion to show that this pilot study has the potential for use in a larger study and is not generalizable in its current form.

“The pilot study indicates the potential impact of the midwifery education app when it is implemented on a larger scale, especially considering that the results show a potential effect on midwives’ learning outcomes.” (p. 32, lines 688–690)

We look forward to hearing from you and would be happy to make further changes, if required.

---

## [Editor Report · Decision Letter 1]

6 Mar 2023

PONE-D-22-03266R1Development and pilot test of a smartphone app for midwifery care in Tanzania: A comparative cross-sectional studyPLOS ONE

Dear Dr. Shimpuku,

Thank you for submitting your manuscript to PLOS ONE. After careful consideration, we feel that it has merit but does not fully meet PLOS ONE’s publication criteria as it currently stands. Therefore, we invite you to submit a revised version of the manuscript that addresses the points raised during the review process.

We look forward to receiving your revised manuscript.

Kind regards,

Michelle L. Munro-Kramer, PhD, CNM, FNP-BC

Academic Editor

PLOS ONE

Journal Requirements:

Additional Editor Comments:

Thank you for the revision. All of the major comments were addressed. There are a few very minor changes before we can accept this for submission:

1) Page 6, line 95 - please move the mHealth definition to the first time the term mHealth is introduced (page 6, line 88)

2) Page 9 - please indicate when this study was conducted (what months and years?)

3) Page 16, line 269 - Please change "giving a maximum score was 10" to "giving a maximum score of 10"

4) Page 24, Line 400-401 - Please change the sentence to read, "To improve the app, they recommended that the reading materials be changed into voice recordings to that they could listen to the contents rather than reading them."

5) Page 24, line 402 - Please change Swahili to Kiswahili for consistency

6) Page 25, line 411 - Please change to "There were no significant differences.."

---

## [Author Response · Author response to Decision Letter 1]

13 Mar 2023

Responses to the Editor 

We express our appreciation to the reviewers for their suggestions for revision, which helped us improve our study. We address your comments with point-by-point responses below, including quotations from the manuscript by page and line number. In the manuscript, we denote changes through tracked changes.

Additional Editor Comments:

Comment: 1) Page 6, line 95 - please move the mHealth definition to the first time the term mHealth is introduced (page 6, line 88)

Response: Thank you very much. We moved the definition to the first time the term mHealth is mentioned (page 6, line 86)

Comment: 2) Page 9 - please indicate when this study was conducted (what months and years?)

Response: We added the study period as “between October 2019 and March 2021.” (page 9, line 139-140)

Comment: 3) Page 16, line 269 - Please change "giving a maximum score was 10" to "giving a maximum score of 10"

Response: We revised as it was suggested. (page 16, line 264)

Comment: 4) Page 24, Line 400-401 - Please change the sentence to read, "To improve the app, they recommended that the reading materials be changed into voice recordings to that they could listen to the contents rather than reading them."

Response: We revised as it was suggested. (page 23, line 389-390)

Comment: 5) Page 24, line 402 - Please change Swahili to Kiswahili for consistency

Response: We revised as it was suggested. (page 24, line 391)

Comment: 6) Page 25, line 411 - Please change to "There were no significant differences.."

Response: We revised as it was suggested. (page 24, line 400)

Thank you very much again.

We look forward to hearing from you and would be happy to make further changes, if required.

---

## [Editor Report · Decision Letter 2]

20 Mar 2023

Development and pilot test of a smartphone app for midwifery care in Tanzania: A comparative cross-sectional study

PONE-D-22-03266R2

Dear Dr. Shimpuku,

We’re pleased to inform you that your manuscript has been judged scientifically suitable for publication and will be formally accepted for publication once it meets all outstanding technical requirements.

Kind regards,

Michelle L. Munro-Kramer, PhD, CNM, FNP-BC

Academic Editor

PLOS ONE

Additional Editor Comments (optional):

Thank you for attending to the most recent changes. I am pleased to accept your manuscript for publication.

---

## [Editor Report · Acceptance letter]

23 Mar 2023

PONE-D-22-03266R2 

Development and pilot test of a smartphone app for midwifery care in Tanzania: A comparative cross-sectional study 

Dear Dr. Shimpuku:

I'm pleased to inform you that your manuscript has been deemed suitable for publication in PLOS ONE. Congratulations! Your manuscript is now with our production department. 

Kind regards, 

on behalf of

Dr. Michelle L. Munro-Kramer 

Academic Editor

PLOS ONE